# Micronutrients to Support Vaccine Immunogenicity and Efficacy

**DOI:** 10.3390/vaccines10040568

**Published:** 2022-04-06

**Authors:** Philip C. Calder, Mette M. Berger, Adrian F. Gombart, Grace A. McComsey, Adrian R. Martineau, Manfred Eggersdorfer

**Affiliations:** 1NIHR Southampton Biomedical Research Centre, University Hospital Southampton NHS Foundation Trust, University of Southampton, Tremona Road, Southampton SO16 6YD, UK; p.c.calder@soton.ac.uk; 2Lausanne University Hospital (CHUV), University of Lausanne, 1011 Lausanne, Switzerland; mette.berger@unil.ch; 3Department of Biochemistry and Biophysics, Linus Pauling Institute, Oregon State University, Corvallis, OR 97331, USA; adrian.gombart@oregonstate.edu; 4University Hospitals of Cleveland, Case Western Reserve University, 11100 Euclid Avenue, Cleveland, OH 44106, USA; grace.mccomsey@uhhospitals.org; 5Centre for Immunobiology, Blizard Institute, Barts and The London School of Medicine and Dentistry, Queen Mary University of London, London E1 2AT, UK; a.martineau@qmul.ac.uk; 6Department of Internal Medicine, University Medical Center Groningen, 9713 GZ Groningen, The Netherlands

**Keywords:** micronutrients, COVID-19 pandemic, vaccine immunogenicity and efficacy

## Abstract

The world has entered the third year of the coronavirus disease 2019 (COVID-19) pandemic. Vaccination is the primary public health strategy to protect against infection with severe acute respiratory syndrome coronavirus 2 (SARS-CoV-2), in addition to other measures, such as mask wearing and social distancing. Vaccination has reduced COVID-19 severity and mortality dramatically. Nevertheless, incidence globally remains high, and certain populations are still at risk for severe outcomes. Additional strategies to support immunity, including potentially enhancing the response to vaccination, are needed. Many vitamins and trace minerals have recognized immunomodulatory actions, and their status and/or supplementation have been reported to correspond to the incidence and severity of infection. Furthermore, a variety of observational and some interventional studies report that adequate micronutrient status or micronutrient supplementation is associated with enhanced vaccine responses, including to COVID-19 vaccination. Such data suggest that micronutrient supplementation may hold the potential to improve vaccine immunogenicity and effectiveness, although additional interventional studies to further strengthen the existing evidence are needed. Positive findings from such research could have important implications for global public health, since deficiencies in several micronutrients that support immune function are prevalent in numerous settings, and supplementation can be implemented safely and inexpensively.

## 1. Introduction

The world has entered the third year of the coronavirus disease 2019 (COVID-19) pandemic. The primary public health strategies to protect against infection with severe acute respiratory syndrome coronavirus 2 (SARS-CoV-2) and the disease it causes (COVID-19) include vaccination, as well as other measures, such as mask wearing and social distancing. Thirty COVID-19 vaccines with full or emergency-use authorizations have been developed globally, with ten approved for use by the World Health Organization (WHO) [1,2]. The vaccines include those based on mRNA technology, non-replicating viral vectors, inactivated virus, or viral protein subunits [1]. While over 10.5 billion vaccine doses have already been administered, the virus has continued to spread rapidly across the world [3]. The omicron variant has proven to be less severe than previous variants, but infection rates globally have increased substantially. This has resulted in record numbers of hospital admissions in some areas and a substantial average daily death rate [3,4]. Beyond COVID-19, other acute respiratory tract infections for which vaccines are available, such as seasonal influenza, also remain major causes of morbidity and mortality. Indeed, the WHO reports that seasonal influenza results in up to 5 million cases of severe illness and 650,000 deaths per year [5].

## 2. Vaccination and COVID-19

Vaccination has reduced the risk of COVID-19 severity and mortality dramatically. The US CDC reports that, in December 2021 and January 2022, unvaccinated US adults were 14 times more likely to die from COVID-19 compared with those who were vaccinated, and 41 times more likely than those who also received a booster dose [6]. Nevertheless, those who are vaccinated, and especially certain vulnerable populations, are still at risk for SARS-CoV-2 infection and severe outcomes. In the US, the incidence of death in those who are 65–79 years of age and fully vaccinated has been comparable to that of unvaccinated 30–49-year-olds, and for those who are at least 80 years old and fully vaccinated, the incidence of death has been comparable to those who are 50–64 years of age and unvaccinated. Similarly, those who are at least 65 years old and have received a booster have a higher mortality rate than those who are 18–49 years old and unvaccinated [6].

Current vaccines appear to show diminished efficacy against common SARS-CoV-2 variants compared to the original viral strain, due to a mismatch between vaccine and viral antigens [7,8]. This problem is expected to persist as additional variants emerge and vaccine antigens continue to differ from those expressed by the virus. In addition, the efficacy of vaccinations decreases over time, as immunity is known to wane, requiring booster doses of the vaccine [9,10]. Finally, vaccination may not be as efficacious in certain vulnerable groups, such as the elderly, as described above. Indeed, older people have diminished antibody responses to vaccines, including the seasonal influenza vaccine and certain COVID-19 vaccines [9,11,12]. Additional strategies to support immunity, including possibly enhancing the response to vaccination to limit the impact of COVID-19, are needed.

## 3. Micronutrient Nutrition and Immunity

The relationship between adequate nutritional status and immune function has been well described [13,14]. Many vitamins and trace minerals are well-known to help ensure an optimal immune response to infection. The key mechanistic and complementary roles that vitamins (e.g., vitamins A, B_6_, B12, C, D, E and K and folate) and trace elements (e.g., zinc, iron, selenium, and copper) play in supporting the innate and adaptive immune responses have been comprehensively reviewed recently [13,14,15,16]. Briefly, preclinical and clinical data indicate roles for specific micronutrients in maintaining the physical barriers in the skin, gastrointestinal, and respiratory tracts (e.g., promoting collagen synthesis and promoting tight junction protein expression); supporting the cells and functions of the innate and inflammatory responses (e.g., oxidative burst, phagocytosis, production of complement proteins and proinflammatory and anti-inflammatory cytokines, and activity of natural killer cells); and supporting the cells and functions of the adaptive immune response (e.g., antigen presentation; T-cell differentiation, proliferation, and function; and B-cell differentiation and antibody production). A mechanistic understanding of vitamin D and immunity is presented below. The reader is directed to the reviews cited above for more details, including reviews related to the functions of specific micronutrients. With the exception of vitamin E, each of these micronutrients has been granted a health claim in the European Union for their role in supporting immune function [17].

The active form of vitamin D, calcitriol (1,25-dihydroxyvitamin D [1,25(OH)2D]), is a potent modulator of the immune system. Immune cells from the lymphoid and myeloid lineages can express the vitamin D receptor (VDR), as well as the enzyme 25-hydroxyvitamin D3-1α-hydroxylase, which allows these cells to convert intracellular calcidiol (25-hydroxyvitamin D [25(OH)D]) to 1,25(OH)_2_D [16,18,19]. This form then binds to the intracellular vitamin D receptor (VDR), which translocates to the nucleus and binds to promoter elements in target genes, thus altering gene expression and profoundly impacting cellular activity [18,19]. Endocrine, paracrine, and intracrine mechanisms of action are considered fundamental ways by which vitamin D impacts immune function [18,19,20]. Thus, when 25(OH)D levels in the blood are insufficient or deficient, immune responses can be limited, potentially leading to increased incidence and severity of disease. Overall, vitamin D positively impacts immunity by supporting barrier function; supporting the differentiation of monocytes to macrophages, as well as the phagocytic and killing capacities of these macrophages; supporting antigen presentation; modulating the inflammatory response, typically by reducing the expression of pro-inflammatory cytokines and increasing the expression of anti-inflammatory cytokines; and impacting antibody production and the activities of various T-cell subsets [13,14,21,22].

Clinically, the most data related to reducing the risk of respiratory infections has been reported for vitamin D. While the results of individual studies are mixed, recent meta-analyses of double-blind randomized placebo-controlled trials indicate that vitamin D supplementation reduces the incidence of acute respiratory tract infections, particularly when supplementation occurs daily [23,24,25,26]. Furthermore, multiple lines of associational clinical evidence indicate that vitamin D inadequacy is associated with increased incidence, severity, and mortality from COVID-19 [27,28,29]. Population-based cohort studies investigating associations between regular use of vitamin D supplements and subsequent risk of COVID-19 have yielded mixed results [30,31,32,33,34], although meta-analyses report protective associations overall [28,35]. Deficiency in other micronutrients is also described to be associated with an increased incidence and/or severity of infectious disease [13,14].

## 4. Micronutrient Nutrition and Vaccine Responses

Vaccination primarily engages adaptive immune responses, although there is growing evidence that innate immunity may also be affected via induction of trained immunity [36]. Based on an extensive body of preclinical and clinical data, it is widely accepted that both innate and adaptive immune responses are supported by an adequate status of a number of vitamins and trace minerals, particularly those named earlier [13,14,37,38,39,40,41]. Furthermore, while not all data are consistent, results from some randomized controlled clinical trials support a cause-and-effect relationship between micronutrient status and the immune response to vaccination. A recent systematic review and meta-analysis of nine clinical studies found lower seroprotection rates in people who were vitamin D deficient compared to those who were adequate, when vaccinated with H3N2 and B strains of seasonal influenza; however, there was no difference in seroprotection against the H1N1 strain [42]. In another study, selenium supplementation in healthy adults was associated with more robust T-cell responses to a live, attenuated polio vaccine, as compared to the unsupplemented group. Supplementation was associated with a more rapid clearance of the attenuated poliovirus vaccine from the body, and virus recovered from the feces contained a lower incidence of genetic mutations [43]. A study in Kenyan infants found that anemia and iron deficiency at the time of vaccination were associated with reduced antibody responses to diphtheria, pneumococcal, and pertussis vaccination. Furthermore, in a follow-up double-blind randomized trial, supplementation with a multi-nutrient powder that included iron improved antibody responses to measles vaccination compared to the multi-nutrient powder lacking iron [44].

Consistent with these data, some recently available studies suggest that adequate vitamin D status or supplementation is associated with the response to COVID-19 vaccination. Chillon et al. explored the relationship between vitamin D status and the IgG response to vaccination with two doses of BNT162b2 in a cohort of 126 healthy healthcare workers (87% women) in Germany [45]. In this study, the authors found no effect of vitamin D status on the IgG response to vaccination through 21 weeks after the second dose [45]. In contrast, data from a longitudinal study of 712 subjects in Greece (mean age 51 years; 62% female) found that replete vitamin D levels were significantly associated with higher antibody titers 3 months post-vaccination with BNT162b2 [46]. Likewise, recently available preliminary data from Jolliffe et al. also demonstrate an independent association between vitamin D supplement use and enhanced humoral responses to COVID-19 vaccination [47]. This study assayed anti-spike antibodies (combined IgG, IgA, and IgM) before and after administration of two doses of ChAdOx1 nCoV-19 or BNT162b2 in 9101 adults (mean age 64 years; 71% female) in a population-based longitudinal study in the UK and examined 66 potential determinants of the antibody response for their possible association with seronegativity. In a fully adjusted multivariable analysis of the vaccine response, regular vitamin D supplementation was associated with a significantly lower risk of post-vaccination seronegativity. While no information was given on dose, frequency of intake, or circulating levels of 25-hydroxyvitamin D, these association data are consistent with a role for vitamin D in supporting vaccine responses, including to COVID-19 vaccines.

The impact of micronutrient supplementation may be even more pronounced in the elderly, who undergo immunosenescence and can suffer from increased rates of infection and poorer response to vaccines [11,37,48]. In one study, healthy individuals 65 years or older who were supplemented with 200 mg/day vitamin E showed more robust cellular immune responses, as well as increased antibody titers to two of three vaccines (hepatitis B and tetanus, but not diphtheria), when compared to those in the placebo group [49]. Consistent with these results, another study investigating this same dose of vitamin E supplementation in healthy elderly men and women showed a significant positive impact on several measures of immune cell function, including chemotaxis and measures of phagocytosis in neutrophils; chemotaxis in lymphocytes; and proliferation, IL-2 production, and cytotoxity in NK cells [50]. Institutionalized elderly patients who received zinc and selenium supplements, either with or without beta carotene, vitamin C, and vitamin E, exhibited higher antibody titers to the influenza vaccine, and a near-significant reduction in respiratory tract infections, whereas supplementation with vitamins alone was associated with lower titers [51]. Finally, a recent study investigated the impact of vitamin D supplementation in healthy vitamin-D-insufficient individuals 65 years or older, who were vaccinated with varicella zoster virus (VZV). Vitamin D supplementation significantly improved secondary antigen-specific cutaneous immune response to VZV challenge, characterized by a reduction of inflammatory monocyte infiltration and an increase in T-cell recruitment to the site of challenge [52].

Attenuation of systemic inflammation is one potential pathway by which supplementation may help improve vaccine effectiveness. Indeed, a chronic high inflammatory state is known to occur in ageing populations, as well as in obesity, HIV infection, and autoimmune diseases. These populations are known to have a suboptimal response to immunizations [53]. It is well documented that vitamin K provision is associated with a reduced production of proinflammatory cytokines, such as interleukin-6 (IL-6),tumor necrosis factor alpha (TNF-α), and interleukin-1 (IL-1) [54,55], which are among the most important cytokines activated during COVID-19, contributing to the cytokine storm in severe COVID-19 patients. Similarly, vitamin D has been shown to decrease systemic inflammation in COVID-19 [56] and in HIV [57,58] and other hyper-inflammatory states. Supplementation with vitamin D and/or vitamin K to decrease systemic inflammation could enhance vaccine effectiveness.

Despite these promising data, it is important to acknowledge that not all studies reveal an improvement in vaccine response as a function of micronutrient supplementation or status. In addition to negative results described above, an intervention study by Provinciale et al. reported no effect of supplemental zinc (400 mg/day for 60 days) on the antibody response to seasonal influenza vaccination in older participants [59]. In addition, while the meta-analysis cited above concluded lower seroprotection rates in people who were vitamin D deficient compared to those who were adequate, the effects are inconsistent with some studies reporting no effect of vitamin D [60,61]. In a study of elderly subjects, there was no association found between levels of vitamin A, vitamin E, or zinc and antibody responses to influenza vaccination. However, it is important to note that, in this study, no participants were deficient in vitamin A or vitamin E, and only 20% had low serum zinc levels [62]. Studies such as these highlight the need for additional intervention trials, as well as the importance of taking into account the baseline intakes and status of their participants.

## 5. Micronutrient Inadequacy and Supplementation

Unfortunately, micronutrient inadequacies are prevalent globally, including inadequacies of those micronutrients that are important for immune function [37,63,64,65,66,67,68,69]. For example, a systematic review of 195 studies, including 168,000 individuals from 44 countries, reported widespread vitamin D inadequacy, with 37% of the studies reporting mean serum 25-hydroxyvitamin D levels lower than the threshold of deficiency (<50 nmol/L) [67]. A systematic review including nearly 250,000 participants from 46 countries indicated that 27% of the American, 80% of the Middle East/African, 62% of the Asian, and 19% of the European adult population had a threshold concentration of α-tocopherol below 20 μmol/L, which is recommended by experts. Furthermore, 13% of the individuals had concentrations below 12 μmol/L, which is the threshold for deficiency [68]. The WHO and the Food and Agriculture Organization of the United Nations (FAO) have reported that iron and vitamin A deficiencies are widespread and of global concern [63,65,66]. Finally, a recent systematic review of elderly adults in 13 Western countries found widespread insufficiencies in several trace elements, including selenium, zinc, iron, and copper [70].

Supplementation should be performed with the goal of reaching an adequate status of the micronutrients in question to achieve beneficial outcomes. The importance of assessing vitamin D status (serum 25(OH)D) after supplementation and achieving adequate vitamin D status to achieve health outcomes has been discussed [71]. As mentioned above, multiple associational studies, systematic reviews, and meta-analyses have concluded that inadequate status of circulating 25(OH)D is associated with higher incidence and severity of COVID-19. These findings have led some authors to conclude that achieving specific thresholds of circulating 25(OH)D (e.g., at least 75 or 125 nmol/L) might reduce the burden of SARS-CoV-2 infection [28,72]. Indeed, in a large population-based cohort study that used public health records, vitamin D supplementation prior to COVID-19 infection was linked to only a modest decrease in infection in the overall population, but also substantial reductions in infection, severe outcomes, and mortality in the subjects that achieved serum 25(OH)D levels of at least 75 nmol/L [30].

## 6. Conclusions

Despite the availability of vaccines, both endemic and pandemic respiratory diseases, such as influenza and COVID-19, lead to extensive morbidity and mortality worldwide. When available, vaccines are, by far, the most important and effective weapons in the arsenal against these and other infectious diseases. Nevertheless, they are not 100% effective, as described above for vaccines against SARS-CoV-2. Similarly, since the 2004/2005 influenza season, the US CDC estimates that influenza vaccination has ranged from 10 to 60% effective for preventing outpatient medical visits due to laboratory-confirmed influenza [73]. Therefore, there is interest in identifying modifiable risk factors for poor immunogenicity and vaccine failure. Given the data presented above, inadequate micronutrient status is worth considering as one of these factors. Unfortunately, nutritional gaps are prevalent in several micronutrients reported to support immune function, including vitamins A, D, and E, as well as iron, zinc, and selenium. Supplementation combining the micronutrients at highest risk of deficit should be considered a safe and effective way to prevent or correct inadequacies, and existing data indicate that such a strategy could support vaccine responses, thereby reducing the incidence and severity of respiratory diseases. On a population level, even an incremental improvement in vaccine immunogenicity could meaningfully reduce the impact of disease and its associated social and economic costs.

## Data Availability

Not applicable.

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
