# Peer review of "Micronutrients to Support Vaccine Immunogenicity and Efficacy"

_vaccines, 2022, doi:10.3390/vaccines10040568_

Round 1

Reviewer 1 Report

This article reviews the importance of micronutrients, such as vitamins and minerals, promoting immunogenicity in infections and vaccines. The revision is focused mainly in SARS-CoV-2 infection and vaccination. The manuscript is well written, but it is difficult to evaluate since it is nor well-structured, therefore it cannot be considered as an original article (stated in the manuscript), communication or review, because its structure is not clear and original data were not included. I suggest defining this article as a (mini-)review, distributing the manuscript into clear sections (Introduction, headings and subheadings) to make more clear to the reader the follow-up of the work.

The gap of knowledge or the originality and contribution of this review is not completely clear. The article is focused on the efficacy, effect or association between micronutrients and immunogenicity in vaccines application or infections, but specific mechanisms involved in immune modulation by micronutrients are not mentioned or explored (Lines 79-83). Some of these mechanisms should be added and discussed. 

I consider that the review topic was not completely covered, since there is a predominance of positive evidence, in terms of the effect of micronutrients. Other works suggesting that micronutrients could have a moderated effect should be also considered and discussed (as performed on lines 118-124 for vitamin D).

Conclusions should also be included.

Author Response

I suggest defining this article as a (mini-)review, distributing the manuscript into clear sections (Introduction, headings and subheadings) to make more clear to the reader the follow-up of the work.

We thank you for the helpful comments and suggestions.  We changed the manuscript into a mini-review format, and distributed the manuscript in sections introduction, four subheadings and conclusions.

Specific mechanisms involved in immune modulation by micronutrients are not mentioned or explored (Lines 79-83). Some of these mechanisms should be added and discussed. 

Thank you for this proposal. We have added on p 2-3 lines 89-121 explanations about preclinical and clinical data supporting the cells and functions of the innate and inflammatory responses and a mechanistic understanding of vitamin D and immunity. The following section is added:

Briefly, preclinical and clinical data indicate roles for specific micronutrients in maintaining the physical barriers in the skin, gastrointestinal, and respiratory tracts (e.g. promoting collagen synthesis, promoting tight junction protein expression); supporting the cells and functions of the innate and inflammatory responses (e.g. oxidative burst, phagocytosis, production of complement proteins and proinflammatory and anti-inflammatory cytokines, activity of natural killer cells); and supporting the cells and functions of the adaptive immune response (e.g. antigen presentation; T cell differentiation, proliferation, and function; B cell differentiation and antibody production). A mechanistic understanding of vitamin D and immunity is presented below. The reader is directed to the reviews cited above for more details, including related to functions of specific micronutrients. With the exception of vitamin E, each of these micronutrients has been granted a health claim in the European Union for their role in supporting immune function [17].

The active form of vitamin D, calcitriol (1,25-dihydroxyvitamin D [1,25(OH)2D]), is a potent modulator of the immune system. Immune cells from the lymphoid and myeloid lineages can express the vitamin D receptor (VDR), as well as the enzyme 25-hydroxyvitamin D3-1α-hydroxylase which allows these cells to convert intracellular calcidiol (25-hydroxyvitamin D [25(OH)D]) to 1,25(OH)2D [16,18,19]. This form then binds to the intracellular vitamin D receptor (VDR), which translocates to the nucleus, binds to promoter elements in target genes, thus altering gene expression and profoundly impacting cellular activity [18,19]. Endocrine, paracrine and intracrine mechanisms of action are considered fundamental ways by which vitamin D impacts immune function [18–20]. Thus, when 25(OH)D levels in the blood are insufficient or deficient, immune responses can be limited, potentially leading to increased incidence and severity of disease. Overall, vitamin D positively impacts immunity by supporting barrier function; supporting the differentiation of monocytes to macrophages, as well as the phagocytic and killing capacities of these macrophages; supporting antigen presentation; modulating the inflammatory response, typically by reducing the expression of pro-inflammatory cytokines and increasing the expression of anti-inflammatory cytokines; and impacting antibody production and the activities of various T cell subsets [13,14,21,22].

Other works suggesting that micronutrients could have a moderated effect should be also considered and discussed (as performed on lines 118-124 for vitamin D).

Thank you for this proposal. We have added on p 5 lines 212-225 a section that not all studies reveal an improvement in vaccine response as a function of micronutrient supplementation or status and added references. The following section is added:

Despite these promising data, it is important to acknowledge that not all studies reveal an improvement in vaccine response as a function of micronutrient supplementation or status. In addition to negative results described above, an intervention study by Provinciale et al. reported no effect of supplemental zinc (400 mg/d for 60 days) on the antibody response to seasonal influenza vaccination in older participants [59]. In addition, while the meta-analysis cited above concluded lower seroprotection rates in people who were vitamin D deficient compared to those who were adequate, the effects are inconsistent with some studies reporting no effect of vitamin D [60,61]. In a study of elderly subjects, there was no association found between levels of vitamin A, vitamin E, or zinc and antibody responses to influenza vaccination. However, it is important to note that in this study no participants were deficient in vitamin A or vitamin E, and only 20% had low serum zinc levels [62]. Studies such as these highlight the need for additional intervention trials, and the importance of taking into account the baseline intakes and status of their participants.

And on p 5-6 lines 247-259

Supplementation should be performed with the goal of reaching adequate status of the micronutrients in question, to achieve beneficial outcomes. The importance of assessing vitamin D status (serum 25(OH)D) after supplementation, and achieving adequate vitamin D status to achieve health outcomes has been discussed [71]. As mentioned above, multiple associational studies, systematic reviews, and meta-analyses have concluded that inadequate status of circulating 25(OH)D is associated with higher incidence and severity of COVID-19. These findings have led some authors to conclude that achieving specific thresholds of circulating 25(OH)D (e.g. at least 75 nmol/L, or 125 nmol/L) might reduce the burden of SARS-CoV-2 infection [28,72]. Indeed, in a large population-based cohort study using public health records, vitamin D supplementation prior to COVID-19 infection was linked to only a modest decrease in infection in the overall population, but substantial reductions in infection, severe outcomes, and mortality in the subjects that achieved serum 25(OH)D levels of at least 75 nmol/L [30].

Conclusions should also be included

Conclusion is now a section p 6 lines 262-279

Despite the availability of vaccines, both endemic and pandemic respiratory diseases such as influenza and COVID-19 lead to extensive morbidity and mortality worldwide. When available, vaccines are by far the most important and effective weapons in the arsenal against these and other infectious diseases. Nevertheless, they are not 100% effective, as described above for vaccines against SARS-CoV-2. Similarly, since the 2004 – 2005 influenza season, the US CDC estimates that influenza vaccination has ranged from 10% - 60% effective for preventing outpatient medical visits due to laboratory-confirmed influenza [73]. Therefore, there is interest in identifying modifiable risk factors for poor immunogenicity and vaccine failure. Given the data presented above, inadequate micronutrient status is worth considering as one of these factors. Unfortunately, nutritional gaps are prevalent in several micronutrients reported to support immune function, including vitamins A, D, and E, as well as iron, zinc and selenium. Supplementation combining the micronutrients at highest risk of deficit should be considered a safe and effective way to prevent or correct inadequacies, and existing data indicate such a strategy could support vaccine responses, thereby reducing the incidence and severity of respiratory diseases. On a population level, even an incremental improvement in vaccine immunogenicity could meaningfully reduce the impact of disease, and its associated social and economic costs.  

Reviewer 2 Report

The article by Dr. Calder and colleagues entitled “Micronutrients to support vaccine immunogenicity and efficacy” is an interesting commentary highlighting the importance of considering micronutrient and vitamin deficiencies in clinical settings to encourage clinicians to start appropriate supplementations aimed at enhancing immune responses to different vaccine types (including anti-COVID-19 vaccines). This is a really important topic, given that anti-COVID-19 vaccines may be not 100% effective with emergence of new SARS-CoV-2 variants.

Some minor revisions are required, as follows:

-An important point to be mentioned is the importance of evaluating the circulating levels of a given micronutrient after its supplementation, in order to assess whether adequate values are achieved. This is particularly true for the immunomodulatory actions of vitamin D: please cite these papers PMID: 34684596 and PMID: 31002167

-Line 37: amend “food fortification”

- SARS-CoV-2 and COVID-19: write them in full the first time you mention in the manuscript: “severe acute respiratory syndrome coronavirus 2”   and “Coronavirus Disease 2019”; then, use abbreviations

-“over 10.5 billion vaccine doses have already been administered”: please provide a reference for this statement

-Line 57: that in December and January 2021

-Line 141: “titers to two different vaccines” please specify the vaccine types

-Line 144 “on several measures of immune cell function” please be more specific

-Lines 157-160: vitamin K deficiency? Also, amend “COVID-19” and write first in full “IL-6, IL-1, TNF-a” (and then abbreviate in brackets).

-Line 162: “COVID infection”: amend into “COVID-19”

-Line 163: “and/or vitamin K”

-Line 193: respiratory diseases

-Line 100: “number of vitamins and trace minerals, particularly those named earlier [13,32,33” at the end of this statement, authors should also quote these papers: PMID: 34282078; PMID: 34533808 ; PMID: 34445700 ; PMID: 35016600  

Author Response

We thank you for the helpful comments and suggestions.

This is a really important topic, given that anti-COVID-19 vaccines may be not 100% effective with emergence of new SARS-CoV-2 variants.

Thank you for your positive comment.

An important point to be mentioned is the importance of evaluating the circulating levels of a given micronutrient after its supplementation, in order to assess whether adequate values are achieved.

We have added the reference

Borsche, L.; Glauner, B.; von Mendel, J. COVID-19 Mortality Risk Correlates Inversely with Vitamin D3 Status, and a Mortality Rate Close to Zero Could Theoretically Be Achieved at 50 Ng/ML 25(OH)D3: Results of a Systematic Review and Meta-Analysis. Nutrients 2021, 13, 3596, doi:10.3390/nu13103596.

-Line 37: amend “food fortification”

We have amended “food fortification”

- SARS-CoV-2 and COVID-19: write them in full the first time you mention in the manuscript: “severe acute respiratory syndrome coronavirus 2”   and “Coronavirus Disease 2019”; then, use abbreviations  

done

-“over 10.5 billion vaccine doses have already been administered”: please provide a reference for this statement

done

-Line 57: that in December and January 2021

Adjusted to December 2021 and January 2022

-Line 141: “titers to two different vaccines” please specify the vaccine types

Thank you, we added on p 4 line 184 the information for vaccine types for hepatitis B and tetanus; the text reads now:

In one study, healthy individuals 65 years or older who were supplemented with 200 mg/day vitamin E showed more robust cellular immune responses, as well as increased antibody titers to two of three vaccines (hepatitis B and tetanus but not diphtheria), when compared to those in the placebo group [49]

-Line 144 “on several measures of immune cell function” please be more specific

We added on p 4 lines 187-189 , several measures of immune cell function, including chemotaxis and measures of phagocytosis in neutrophils; chemotaxis in lymphocytes; and proliferation, IL-2 production, and cytotoxity in NK cells.

-Lines 157-160: vitamin K deficiency? Also, amend “COVID-19” and write first in full “IL-6, IL-1, TNF-a” (and then abbreviate in brackets).

Done p 5 lines 205-206

-Line 162: “COVID infection”: amend into “COVID-19”

done p 5 line 209

-Line 163: “and/or vitamin K”

done p 5 line 210

-Line 193: respiratory diseases

done p 6 line 262

-Line 100: “number of vitamins and trace minerals, particularly those named earlier [13,32,33” at the end of this statement, authors should also quote these papers: PMID: 34282078; PMID: 34533808 ; PMID: 34445700 ; PMID: 35016600  

Thank you for this information, on p 3 line 141 we added the three references which refer to the role of micronutrients in vaccine efficacy.

  1. Chiu, S.-K.; Tsai, K.-W.; Wu, C.-C.; Zheng, C.-M.; Yang, C.-H.; Hu, W.-C.; Hou, Y.-C.; Lu, K.-C.; Chao, Y.-C. Putative Role of Vitamin D for COVID-19 Vaccination. IJMS 2021, 22, 8988, doi:10.3390/ijms22168988.
  2. Velikova, T.; Fabbri, A.; Infante, M. The Role of Vitamin D as a Potential Adjuvant for COVID-19 Vaccines. Eur Rev Med Pharmacol Sci 2021, 25, 5323–5327.
  3. Lai, Y.-J.; Chang, H.-S.; Yang, Y.-P.; Lin, T.-W.; Lai, W.-Y.; Lin, Y.-Y.; Chang, C.-C. The Role of Micronutrient and Immunomodulation Effect in the Vaccine Era of COVID-19. Journal of the Chinese Medical Association 2021, 84, 821–826, doi:10.1097/JCMA.0000000000000587.

Round 2

Reviewer 1 Report

The article improved substantially. I recommend publishing it in its current state.